# Recent Advances in the Management of Rosacea through Natural Compounds

**DOI:** 10.3390/ph17020212

**Published:** 2024-02-06

**Authors:** Iulia Semenescu, Diana Similie, Zorita Diaconeasa, Corina Danciu

**Affiliations:** 1Department of Pharmacognosy, “Victor Babeș” University of Medicine and Pharmacy, Eftimie Murgu Square, No. 2, 300041 Timisoara, Romania; iulia.semenescu@umft.ro (I.S.); corina.danciu@umft.ro (C.D.); 2Research Center for Pharmaco-Toxicological Evaluation, “Victor Babeș” University of Medicine and Pharmacy, Eftimie Murgu Square, No. 2, 300041 Timisoara, Romania; 3Department of Food Science and Technology, Faculty of Food Science and Technology, University of Agricultural Science and Veterinary Medicine, Calea Manastur, 3-5, 400372 Cluj-Napoca, Romania; zorita.sconta@usamvcluj.ro

**Keywords:** rosacea, natural compounds, antierythematous, anti-inflammatory, antioxidant, antimicrobial

## Abstract

Rosacea is a chronic skin disorder that affects more than 5% of the world’s population, with the number increasing every year. Moreover, studies show that one-third of those suffering from rosacea report a degree of depression and are less compliant with treatment. Despite being the subject of prolonged studies, the pathogenesis of rosacea remains controversial and elusive. Since most medications used for the management of this pathology have side effects or simply do not yield the necessary results, many patients lose trust in the treatment and drop it altogether. Thus, dermato-cosmetic products with natural ingredients are gaining more and more notoriety in front of synthetic ones, due to the multiple benefits and the reduced number and intensity of side effects. This review is a comprehensive up-to-date report of studies that managed to prove the beneficial effects of different botanicals that may be useful in the short and long-term management of rosacea-affected skin. Based on recent preclinical and clinical studies, this review describes the mechanisms of action of a large array of phytochemicals responsible for alleviating the clinical symptomatology of the disease. This is useful in further aiding and better comprehending the way plant-based products may help in managing this complex condition, paving the way for research in this area of study.

## 1. Introduction

Rosacea is a chronic, inflammatory skin condition affecting predominantly facial convexities such as the forehead, nose, cheeks, and chin [1]. It is a chronic condition, affecting more than 5% of the world’s population [2]. A major difference between the sexes is the fact that the symptoms of rosacea appear earlier in women, around middle age (30–50 years) [1,3]. Also, men are more prone to develop rhinophyma [3]. Ethnicity also influences the incidence of rosacea, with people with fair skin being affected the most, followed by Asians and people with dark skin [4].

In 2002, the National Rosacea Society Expert Committee (NRSEC) constituted a report developing a practical classification and staging for rosacea [5]. This classification has been used in practice and helped many healthcare professionals to properly diagnose and treat rosacea-affected skin. In 2017, the committee reunited for an updated version of this classification, based on a better understanding of the disorder [6]. The review proposes a slightly different classification, based on phenotypes, and provides clearer parameters and guidelines [6,7]. Unlike the initial classification, this updated one takes into consideration the simultaneous occurrence of more than one subtype, and possibly the progression of the disorder, from one subtype to the other. The 2017 classification is based on the fact that rosacea can manifest through a multitude of combinations of symptoms, and is also in agreement with the 2016 diagnosis criteria recommended by the global rosacea consensus panel. The 2017 classification is currently used in practice by most physicians. Table 1 reunites the classification of the NRSEC with a few other perspectives on classification criteria, so it can provide a complete and complex image of the way rosacea is currently viewed and treated [5,6,7,8].

Nevertheless, the initial clinical subtypes brought forward in 2002 are still helpful and used by practitioners because they help identify a treatment direction and give the patient a better understanding of the disorder (Table 2).

The most common subtype of rosacea is ETR, affecting over 50% of the total cases of the disease, followed by the papulopustular subtype, with a percentile of around 30% of men and 35% of women being affected [13]. Based on the above-mentioned statistic, it is not surprising that most treatments and most studies generally address these manifestations. Also, the mechanisms presented in this review, as well as the natural approaches, predominantly address these two subtypes. Although not all patients experience the same symptoms, central facial erythema, and telangiectasia are usually common throughout the first three subtypes. Ocular rosacea is sometimes correlated with the symptoms affecting the skin, but other times it presents as the “only” symptom, thus making it hard to diagnose [12].

There are a number of different topical and oral therapies that may be administered in rosacea. Some of them are FDA-approved and include metronidazole, ivermectin, azelaic acid, and oral doxycycline in under-microbial dosage [14]. However, most of the medication, either FDA-approved or not, has many side effects, and patients tend to drop the treatment before it has time to manifest the good effects on their skin. This is one of the reasons why it seems like more and more doctors, and patients alike, turn to botanical formulations.

The use of herbs as medical and health sources dates back more than 3000 years. In traditional Chinese Medicine, plants were used to maintain health and help the body repair disrupted balance in order for it to be able to fight disease and heal [15]. In India, the book of Ayurveda describes over 2000 plants and their medicinal potential [16]. The WHO estimates that almost 80% of the world population is still using traditional systems of medicine for their primary health care [17]. These traditional systems of medicine have always been important since many of the modern drugs (e.g., aspirin, morphine, ephedrine, atropine, digoxin) are derived from plants used traditionally by indigenous people [18]. The natural products’ research and investigation have led, on many occasions, to the development of new structures for drugs, thus helping to treat diseases from all areas. However, only a small percentage of today’s known plants have been clinically investigated for their potential medical use: 6% for biological activity and 15% for their phytochemical composition [18]. There is an acute need for more studies on plants, botanical extracts, and therapies, so that the medical fields and patients alike may benefit from their huge potential.

The purpose of this review is to consolidate the findings from preclinical and clinical studies that demonstrated the efficacy of herbal extracts and phytocompounds in managing rosacea. It provides a resource for healthcare practitioners and researchers interested in these compounds as alternative or complementary therapies.

## 2. Methods

The objective of this review is based on the need for a structured collection of information on the use of herbal products in the treatment of rosacea. Aspects of the physiopathology of rosacea and conventional treatments as well as the main plants and plant extracts with therapeutic potential have been gathered together.

A search was performed in the scientific databases Springer, MDPI, PubMed, and Google Scholar, from February 2023 until September 2023. Articles were searched in the databases mentioned above by combining keywords with the Boolean operators OR and AND. For example, the following search string was used: (“rosacea treatment”) OR (“treatment of rosacea”) AND (“rosacea pathogenesis”) OR (“physiopathology of rosacea”). References of articles of interest were also explored to detect other relevant information.

The figures are original and were designed using Microsoft PowerPoint (version 2010) and Canva (www.canva.com, accessed from 10 June 2023 to 30 January 2024).

## 3. Pathophysiology and Mechanisms in Rosacea

The pathophysiology of rosacea remains unclear and controversial. It seems that the clinical and histopathological findings point toward the involvement of various inflammatory and immune-mediated processes. Recent data suggest that alteration in the immune response, genetic factors, vascular dysfunctions, oxidative stress, neurogenic inflammation, and *Demodex* and/or *Helicobacter pylori* infection are possible causes of the disease [1,3,4]. There are several factors that appear to have main contributions to its development [19,20,21,22]: I. abnormal immunity; II. dermal matrix degeneration and endothelial damage; and III. blood vessels, and neurovascular and vascular origins (Figure 1).

Erythema and flushing, which are the most common incipient symptoms of rosacea, result from increased blood flow of the skin surface vessels and increased vessel density at the skin surface [12]. Recent studies, however, have found a link between the (initial) erythema and vascular manifestations and neurovascular dysregulation combined with innate immune response [19]. Abnormal cathelicidin peptides seem to be produced by an early and exaggerated immunological response, which also stimulates and regulates leukocytic chemotaxis, angiogenesis, and the expression of extracellular matrix proteins [10]. Aberration in cutaneous vascular homeostasis has also been suggested to play an important role. Reactive oxygen species (ROS)-induced damage to keratinocytes, fibroblasts, and endothelial cells by the release of interleukin (IL)-1 and tumor necrosis factor (TNF)-α have been depicted. Elevated levels of various proteases like serine protease kallikrein 5 (KLK5) have been also reported in rosacea [10].

Another pathological mechanism of rosacea, especially in the first two stages of the disease, is the excessive activation of Toll-like 2 receptors at the level of keratinocytes, involved in the immune response [25]. The activation of these receptors triggers a series of processes caused by cytokines and chemokines, which have a pro-inflammatory role [25]. These compounds are responsible for the appearance of erythema and telangiectasia [25]. It was also demonstrated that CD4^+^ T cells are involved in the pathophysiology of rosacea [25]. The activation of TLR-2 receptors leads to the transcription of NF-κB, which further increases the transcription of cathelicidin-37 and kallikrein-5/7 [26]. Cathelicidin and kallikrein are involved in angiogenesis, inflammation, and chemotaxis [26]. Kallikrein-5 inhibitors represent a new category of therapeutic agents that can bring numerous benefits in understanding the pathophysiology of rosacea [27].

Telangiectasia is also the result of a vascular dysfunction that can be justified by overexpression of vascular endothelial growth factor (VEGF) after UV-B radiation exposure [26].

*Demodex folliculorum* and *Helicobacter pylori* may also be responsible, as triggers or causes, for the appearance of rosacea [28]. It may also develop as a manifestation of systemic diseases. Smoking, obesity, and even gut-associated diseases have significant risks for the development of rosacea. In addition, other metabolic, psychiatric or neurologic disorders, certain drugs, and lifestyle conditions also show a significant contribution to the disease [29].

For the erythematotelangiectatic and papulopustular subtypes, the classical treatment includes topical agents, such as azelaic acid, metronidazole, or sodium sulfacetamide–sulfur [30]. In some cases, patients are prescribed an oral antibiotic [10]. In telangiectasias and persistent erythema, nonablative lasers, vascular lasers, or intense pulse light therapy are common. Nonetheless, the phymatous version must include medical therapy such as isotretinoin and, if the condition advances, interventions such as microdermabrasion or carbon dioxide laser might be necessary [10].

One very important aspect is that this condition, no matter the type, will not spontaneously resolve, and a proper skin care regimen is essential. The patient also needs to avoid common triggers. Several external factors can aggravate the symptoms of rosacea, and they include ingestion of spicy foods, caffeine or alcohol, exposure to heat and UV rays, and emotional stress [1,11]. Many authors believe that most of the triggers are stimulating inflammatory cascades and increase ROS [5,12,13,31]. The National Rosacea Society ran a survey on 1066 rosacea patients to find the most common rosacea triggers [32]. Figure 2 represents the top 10 triggers found in the survey, based on how many respondents selected each answer.

Because of the incomplete understanding of the pathogenesis of rosacea, overall treatment can be difficult, with most medications having many side effects that counteract the benefits [33]. This makes it even harder for health providers to approach the condition and the patient positively. And it makes it even more difficult for the patient to be compliant long-term, resulting in the treatment not being adhered to long enough to provide effective results.

Herbal medicines offer a range of benefits, one of which is their complex compositions. The intricate chemical makeup of these medicines allows for the possibility of biological activity resulting from the interaction of multiple compounds [34]. Such synergistic effects can lead to unique outcomes that cannot be replicated by single compounds. Therefore, the study of herbal medicines can lead to the development of more effective and efficient treatments.

In recent years, scientists have looked into the benefits that different botanical extracts may bring to those suffering from rosacea–erythematotelangiectatic and papulopustular subtypes. Most botanicals found to have positive effects in treating rosacea have important anti-inflammatory and anti-oxidant components, such as polyphenols (flavonoids, anthocyanins, and phenolic acids), licochalcones, carotenoids, trace elements, vitamins, etc. [35].

Following this perspective, it makes sense that an herbal extract may have more than one positive effect on the skin. Moreover, the risk–benefit balance favors the benefits, since most botanicals are associated with a lower risk of developing adverse reactions.

## 4. Standard Management and Treatment Options for Rosacea

Even though technology is becoming more advanced every day, there is still no treatment that can cure rosacea. All the treatment options are mostly addressing the symptoms, helping to reduce or control the disease up to a point. There are a number of therapies that have been approved by the FDA. In 2020, the *Journal of the American Academy of Dermatology* published a new standard in management options for rosacea [14]. It was developed via the consensus of an expert committee and review panel consisting of 27 rosacea experts throughout the world. Considering the changes in the rosacea classification that took place in 2017 [8], the treatment options and management of rosacea also needed an update. According to the FDA, the topical therapy for papules/pustules and inflammatory rosacea may include azelaic acid 15% (applied twice daily), metronidazole 0.75% or 1%, ivermectin cream 1% (applied once daily), and 10% sodium sulfacetamide [14]. Patients with more severe rosacea or those suffering from ocular rosacea may also need oral therapy. Oral administration includes doxycycline, in the form of modified release (30 mg immediate release and 10 mg delayed release) [14]. Because of the lower-than-regular dose and modified release of doxycycline, there are fewer adverse reactions and the development of bacterial resistance is minimal. Other rosacea therapies for systemic administration include oral metronidazole or isotretinoin [1]. However, systemic medication, as well as topical ones, have risks of side effects when used over a longer period [33].

Because of the complex and not yet clarified physiopathology of rosacea, many dermatologists initially recommend combination treatments, usually followed by single therapy, to maintain the results [14]. Some of these combinations include topical metronidazole with oral doxycycline, topical brimonidine, topical ivermectin, or topical ivermectin with oral doxycycline [7]. However, there seems to be limited evidence supporting the efficacy of combination treatments and even less evidence that they would reduce the side effects [14,36].

Besides the standard treatment options presented, dermatologists recommend lifestyle adjustments to avoid remissions and keep the disease under control. Some of the most important changes to be taken into consideration are the exclusion of common triggers, such as stress, hot baths, extremely hot/cold weather, spicy food, and most importantly sun exposure [37].

Skincare is also an important factor to be taken into consideration when caring for rosacea-affected skin. There is a considerable amount of skincare products claiming to reduce the appearance of redness and soothe the skin. Clinical data to support these claims are rather scarce. However, dermatologists do recommend gentle cleansers, calming moisturizers (usually containing natural ingredients such as panthenol, licorice root, or aloe vera), and sunscreen [34].

## 5. Natural Compounds for the Management of Rosacea

Since most medications have plenty of side effects or simply do not yield the necessary results, many patients lose trust in the treatment and drop it altogether. Rosacea is a skin condition that requires long-term treatment and a proper skincare regimen in order to obtain the best results. This is why dermatologists and patients alike are turning to alternative products—products that are overall gentler to the skin. Most of these products contain natural ingredients. However, a good approach would be for a combination of the medicated treatments and natural products—that way, the side effects are a lot more tolerable, and the patient benefits from the extra benefits of the botanical extracts.

Botanicals have been used worldwide for centuries in skincare as wound healing and anti-aging remedies. They provide alternatives for doctors and patients when standard prescription medications have too many side effects and the patients are not compliant. Additionally, they can be used to minimize the negative effects of prescription medications or to improve their therapeutic effects. Plant extracts contain compounds from different chemical classes that due to their synergism have an essential role in the management of rosacea.

It is important that people become more informed about botanical remedies, as the popularity and use of herbal treatments are constantly increasing [38]. As already mentioned, there are many studies attesting to the beneficial use of botanicals, but dermatologists and patients alike should also take into consideration that they are not completely without adverse effects. However, they are much milder and less severe than in the case of classic medication. Most of these reactions consist of transient burning or pruritus [31].

Below are some examples of natural compounds that are used or may be used for the management of rosacea (Table 3). Even if some of them have not been directly studied in regard to rosacea, they have been shown to address symptoms that are also common in rosacea (mostly anti-inflammatory, antioxidant, and vasoprotective). This is why these compounds may be further investigated for possibly targeting the first two primary features of rosacea (erythema and papules).

All of the plants presented in this table contain a huge number of active components. Owing to the combination and complexity of their components, they manage to express a unique synergy that is able to address at least one (usually more) of the factors responsible for triggering rosacea (Figure 3).

### 5.1. Glycyrrhiza glabra L. and Glycyrrhiza inflata L. (Licorice)

Licorice has long been used in alternative medicine to treat various inflammatory conditions due to its complex botanical composition. *G. glabra* is known to be especially rich in flavonoids and saponins (glabridin, glychyrrizic acid, glycyrrhizin, lichochalcone, and liquiritigenin), but also polysaccharides and other phenols [61]. Some of the most studied compounds in *G. glabra* are glabridin and glycyrrhizin, both of which are found in higher quantities and have anti-irritant and anti-inflammatory properties [62,63]. Because it inhibits the generation of superoxide anion and cyclooxygenase activity, licorice has been demonstrated to have anti-inflammatory properties [64].

In 2019, Frattaruolo et al. conducted a complex study that showed that *G. glabra* leaf extract (obtained either through maceration or ultrasound-assisted methods) has high antioxidant and anti-inflammatory abilities, inhibiting LPS-induced expression of TNF-α, IL-1, and IL-6 [62]. The research group isolated three compounds from the extract and followed the experiment with the one that showed the highest antioxidant potential—licoflavone. This compound (licoflavone), as well as the whole phytocomplex from which it is extracted, was able to decrease iNOS and COX2 expression, making it a potential ingredient in skincare addressing sensitive and inflamed skin, such as rosacea [62].

M. Saeedi et al. conducted another study that evaluated the effect of *G. glabra* (standardized in glycyrrhizinic acid) on atopic dermatitis [65]. Topical preparations of 1% and 2% (using propylene glycol as a co-solvent and Carbopol 940 as a gelling agent) were studied during a double-blind clinical trial in comparison with a base gel. Over the course of two weeks, a two percent licorice topical gel considerably reduced the scores for erythema, edema, and itching, making it a great option for skin affected by atopic dermatitis [65]. Taking into consideration that rosacea is also characterized by sensitive skin, erythema, inflammation, and itching, it may be concluded that the same effects will concur on rosacea-affected skin, thus making *G. glabra* extract a great candidate for this skin condition.

*G.inflata* has a similar composition to *G. glabra*, namely flavonoids and saponins (glabridin, glychyrrizic acid, glycyrrhizin, lichochalcone, and liquiritigenin), but contains a higher amount of licochalcone A [66].

In a study examining the benefits of skin care using licochalcone A (Eucerin Redness Relief; Beiersdorf, Inc., Hamburg, Germany), a retrochalcone derived from *Glycyrrhiza inflata* L., patients with mild-to-moderate facial redness showed improvements in mean erythema and quality-of-life scores at 4 and 8 weeks. The outcomes showed similar efficacy to azelaic acid and topical metronidazole [67]. Another study that involved healthy individuals observed that applying an extract containing licochalcone A twice a day for three days significantly decreased the amount of erythema caused by shaving and UV exposure when compared with a vehicle control [64].

### 5.2. Chrysanthemum indicum L. (Indian Chrysanthemum)

It has been shown that vascular changes may play a major role in rosacea pathogenesis. The extract of *C. indicum* has been shown to contain a unique combination of saponins, phenylpropenoic acids, and flavonoids. Owing to this complex composition, it has a high improvement rate in vascular wall permeability, being well documented for increasing the mechanical resistance of capillaries [41].

Rigopoulos et al. conducted a randomized, double-blind, parallel-group, placebo-controlled, multicenter trial comparing the efficacy of a 1% *C. indicum* cream (*n* = 125) with a placebo (*n* = 121). For 12 weeks, patients were asked to apply the product twice a day [68]. Treatment with *C*. *indicum* extract cream resulted in a significant (*p* < 0.05) improvement in erythema severity, and overall rosacea severity, compared with baseline and placebo values. The adverse reactions noted in the study were few and mild, with no significant difference between the *C. indicum* extract study group and the placebo group.

### 5.3. Avena sativa L. (Colloidal Oat)

Colloidal oat has been used for centuries as a soothing agent to relieve itch and irritation associated with various xerotic dermatoses [69]. Its capacity to calm and shield irritated skin contributes to positive results. Its phytochemical composition includes a variety of active principles, such as polysaccharides, proteins, lipids, saponins, enzymes, flavonoids, vitamins, and polyphenols (avenanthramides—AVA) [70]. Colloidal oatmeal was approved as a secure and reliable skin protectant by the FDA in 1989. In 2003, colloidal oatmeal became an approved over-the-counter monograph ingredient [70].

Most authors consider that the AVAs in oats are actually responsible for the beneficial effects attributed to oats or colloidal oats. Researchers have demonstrated anti-inflammatory and antipruritic properties by decreased production of NF-kB in keratinocytes and reduced proinflammatory cytokine (e.g., IL-8) production [71]. Based on these findings, other studies have looked into the beneficial properties of colloidal oatmeal (anti-inflammatory, hydrating, and antipruritic) for the management of common inflammatory dermatoses, such as atopic dermatitis. Taking into consideration that atopic dermatitis and rosacea have quite a few things in common (redness, inflammation, itching, sensitivity to UV light), the benefits of oatmeal can extend to rosacea-affected skin, too.

Oats have important antioxidant, ultraviolet (UV)-absorbent, and anti-inflammatory properties attributed to the ferulic, caffeic, and coumaric acids, as well as to the flavonoids and tocopherol (vitamin E) [72]. Moreover, the saponin content in oatmeal helps to solubilize dirt, oil, and sebaceous secretions, normalizing the skin’s pH and relieving sensitivity [73].

In 2016, Southall M. et al. conducted an investigator-blinded randomized clinical study in order to demonstrate the effectiveness of an oatmeal lotion in improving the skin barrier and moisture [74]. After five weeks, clinical evaluations of skin dryness showed significant improvements (*p* < 0.05) at all time points during the treatment. Even 2 weeks after the last application, the skin was significantly better when compared with baseline values. The same study showed that colloidal oatmeal decreased TEER in the skin by 52% compared with the untreated tissue. Taking into consideration that pro-inflammatory cells are considered partially responsible for some of the rosacea manifestations, these results are of great value for developing future botanical treatments for rosacea.

### 5.4. Quassia amara L. (Bitter Wood)

*Quassia amara* is a small, tropical tree, also known as bitter wood. It contains a number of important phytochemicals, such as bitter oils, steroids, and triterpenes [75]. There are various biological activities described in the specialized literature; the anti-inflammatory properties are just one of the reasons why *Q. amara* wood extract has been chosen to be studied for rosacea treatment.

A group of 30 patients with various grades of rosacea (I–IV) were investigated in a single-center, open-label study. They were treated with a topical gel with 4% *Quassia amara* extract for 6 weeks. The evaluation took into consideration the main symptoms of rosacea: flushing, erythema, telangiectasia, and papule and pustule scores. The study showed an improvement for most participants, with a significant effect on telangiectasia [76].

During the same study, the anti-inflammatory effect of quassia extract applied to the skin was tested. Among the screened pro-inflammatory mediators, quassia extract displayed inhibitory activity on TNF-a’, IL-1ß, and, most interestingly, nitric oxide production, the increased production of which is known to increase vasodilation [76].

### 5.5. Tenacetum parthenium L. (Feverfew)

Feverfew (*Tenaceetum parthenium* L.), a member of the *Asteraceae* family, is a medicinal herb used traditionally to reduce fever and treat headaches, arthritis, and digestive problems [77]. It has potent anti-inflammatory, antioxidant, and anti-irritant properties. Its main components are volatile oils (L-camphor, *trans*-chrisantenyl acetate), flavonoids, and sesquiterpene lactones (parthenolides) [46]. Feverfew inhibits 5-lipoxygenase and cyclooxygenase, resulting in a reduction in platelet aggregation. Parthenolides, especially, inhibit serotonin release from platelets [78].

The strong irritating properties of parthenolides have restricted the topical application of feverfew. Nonetheless, an industry-patented product that makes parthenolides removable was created. As a result, feverfew PFE (Aveeno; Johnson and Johnson Consumer Companies, Inc., New Brunswick, NJ, USA), a purified feverfew extract, was developed with the capacity to minimize skin irritation and redness on the face by preventing the release of inflammatory markers from activated lymphocytes and reducing neutrophil chemotaxis [79,80]. There is evidence that feverfew PFE exhibits antioxidative and anti-inflammatory properties by (1) inhibiting proinflammatory mediators released from macrophages (nitric oxide, PGE2, and tumor necrosis factor-alpha) and human blood monocytes (tumor necrosis factor-alpha, interleukin [IL]-2, IL-4, and interferon); (2) decreasing neutrophil chemotaxis; (3) reducing NF-kB-dependent gene transcription; and (4) inhibiting the release of IL-8 and adhesion molecules expressed from keratinocytes [80,81]. The group of Sur et al. evaluated a solution of 1% PD-Feverfew, using 70% ethanol/30% propylene glycol as the vehicle. A small amount of the solution was applied to reconstituted human epidermis 2 h before exposure to UV light. After UV irradiation, there was a 60% reduction in IL-1alfa release in the skin cells treated with PD-Feverfew than in the placebo-treated control skin equivalents [81]. After being investigated, it was found that this purified extract reduced irritation from shaving and had protective properties against UV exposure and skin irritation. It also improved facial flushing, blotchiness, and tactile roughness [81,82,83].

### 5.6. Artemisia annua L. (Wormwood)

Artemisinin, isolated from *Artemisia annua* L., has been used for over 30 years as an antimalarial agent [84]. Used by over a million patients, artemisinin has demonstrated its effectiveness and therapeutic safety [25]. Recently, some animal and clinical studies have shown that the active principles of this plant are useful in the treatment of rosacea [25,84]. Experimental studies have shown that artemisinin has anti-angiogenic and anti-inflammatory effects, which are extremely useful in the treatment of rosacea [25].

In a study conducted on mouse models, Yuan et al. demonstrated that artemisinin reduces the infiltration of CD4^+^ T cells, neutrophils, and macrophages, inhibits inflammation via the NF-kB signaling pathway, and suppresses angiogenesis [25]. Before being administered, artemisinin was diluted with filtered DMSO and afterward introduced into the food of laboratory animals [25].

Wang et al. evaluated the efficacy of artemether, a lipid-based derivative of artemisinin, compared with metronidazole in humans [84]. Patients in the first study group were administered topical artemether emulsion (1%), and those in the second group were administered topical metronidazole emulsion (3%) for 4 weeks [84]. The conclusion of this study was that patients treated with artemether emulsion felt a more significant improvement in the symptoms of rosacea, and the beneficial effects were maintained during and after 8 weeks [84]. These beneficial effects of artemether are due to its strong anti-inflammatory action, but also to its anti-*Demodex folliculorum* activity [85]. There are also data suggesting that the efficacy of artemether is similar to that of doxycycline hydrochloride in patients with rosacea [84].

Another compound of interest is artesunate, one of the bioactive derivatives of artemisinin [86]. It is an innovative, effective, and safe antimalarial agent that has been shown to have antiparasitic and antiangiogenic actions [86]. Artesunate improves the symptoms of rosacea due to the fact that it inhibits the JAK-STAT3 signaling pathway, which is involved in proliferation, apoptosis, differentiation, and the release of pro-inflammatory cytokines [86]. In conclusion, artesunate is an alternative treatment method for rosacea, with potential action similar to doxycycline and with significantly reduced side effects [86].

### 5.7. Matricaria recutita L. (Chamomile)

Chamomile (*Matricaria recutita* and *Chamaemelum nobile*) has a wide range of beneficial active components like bisabolol, chamazulene, apigenin, and quercetin, which are mostly present in its volatile oils [51]. The complex composition makes the extract able to inhibit cyclooxygenase and lipoxygenase as well as regulate the T helper cell (Th2) activation and histamine release [85,87]. Recent studies have shown that topical applications are beneficial in atopic dermatitis and skin irritation. One study concluded that the anti-inflammatory effect in the case of medium-degree eczema is comparable to the effect produced by hydrocortisone 0.25% [88,89]. Symptoms (also present in rosacea) such as itching, erythema, and scaling were significantly reduced.

Even though it has soothing effects, chamomile may potentially induce allergic contact dermatitis, so caution is warranted when used on sensitive skin.

### 5.8. Potentilla erecta L. (Tormentil)

Clinical data show that tormentil extract has a vasoconstrictor effect and could be used to improve telangiectasia in patients with rosacea [34]. This vasoconstrictor effect can be partially attributed to the radical capture of NO, which has a vasodilatory role at the dermal level, and through the inhibition of the endothelial NO-synthase (eNOS) [34].

Kaltalioglu et al. conducted an in vivo study to address the antioxidant, antimicrobial, and wound-healing properties of *P. erecta* methanolic extract in diabetic rats. Besides other important findings of the study, it showed that *P. erecta* managed to statistically increase skin collagen and decrease oxidative events [90].

### 5.9. Camellia sinensis Kuntze (Green Tea)

The leaves and buds of the tea plant (*Camellia sinensis* Kuntze) contain a significant number of antioxidant, anti-inflammatory, and photoprotective polyphenols known as catechins [91]. Epigallocatechin gallate and epicatechin gallate are two of the most studied polyphenols (but not the only studied polyphenols). They have a powerful ability to initiate cellular/molecular responses in the epidermis, thus attracting a lot of interest in the field of dermatology [92]. The antioxidant activity is manifested by eliminating reactive oxygen species (ROS) and inhibiting nitric oxide synthetase. Moreover, the inhibition of lipooxygenase and cyclooxygenase makes up for both antioxidant and anti-inflammatory effects.

There are a number of in vivo [91] and in vitro [93,94] studies that show the ability of green tea extract to absorb UV radiation. This in itself makes the green tea extract particularly useful in the treatment of rosacea because UV radiation is one of the triggers of the disease. The application of topical green tea formulations (containing epigallocatechin gallate and epicatechin gallate) has been shown to decrease UV-induced erythema and to reduce DNA damage as demonstrated by measuring cyclobutane pyrimidine dimers [95].

There are also some studies that show that these types of proanthocyanidins have a beneficial effect on rosacea by suppressing the expression of the vascular endothelial growth factor and the hypoxia-induced factor (HIF-1), thus inhibiting the angiogenesis process [34].

### 5.10. Artemisia lavandulaefolia L.

The whole extract, essential oils, and secondary metabolites of these plants show antimicrobial, antifungal, insecticidal, anti-inflammatory, and anti-angiogenic action [27,54].

Roh et al. evaluated the effect of two bioactive constituents of this plant: 3,5-dicaffeoylquinic acid (isochlorogenic acid A) and 4,5-dicaffeoylquinic acid (isochlorogenic acid C) [27]. They demonstrated that chlorogenic acids A and C inhibit KLK5 protease activity, resulting in a reduction in the level of active cathecyclidine [27]. This mechanism of action is responsible for suppressing the inflammatory response mediated by macrophages and mast cells and for inhibiting the proliferation and migration of vascular endothelial cells [27]. Thus, these compounds have beneficial effects on the inflammatory response and erythema in rosacea patients [27]. The same study also demonstrated that 3,5-caffeoylquinic acid extracted from *A. lavandulaefolia* L. has a protective effect against UVA radiation, further making it useful in the management of rosacea [27,96].

### 5.11. Citrus junos Tanaka (Yuzu Citrus Fruit)

Citrus fruits are well known for their nutrition and health benefits. Yuzu (*Citrus junos* Tanaka) is common in Asian countries, especially in Korea and Japan. It is substantially rich in antioxidants and has been demonstrated to have great anti-inflammatory action for the skin [97]. These effects (antioxidant and anti-inflammatory) are mostly attributed to the phenols and flavonoids contained in the peel/seeds of Yuzu fruit. Hirota et al. showed that limonene reduces reactive oxygen species production and the activity of activated B cells (NF-kB), resulting in both antioxidant and anti-inflammatory properties [98].

In a study developed in 2018 (Citron Essential Oils Alleviate the Mediators Related to Rosacea Pathophysiology in Epidermal Keratinocytes), it was demonstrated that citron essential oils have a suppressive effect on LL-37, KLK5, and other pro-inflammatory mediators involved in the pathophysiology of rosacea [99]. Due to this study, it can now be concluded that citron essential oils may actually normalize the innate immune system response in rosacea patients, and help improve the symptoms in this skin condition.

### 5.12. Achillea millefolium L. (Yarrow)

Yarrow is a well-known plant that contains an important number of phytochemicals, such as flavonoids, phenols, phytosterols, terpene, and essential oils [57,100]. One of the most important benefits of Yarrow extract is its anti-inflammatory properties. The polysaccharide fraction Am-25-d, obtained from the aqueous extract of *A. millefolium*, increases lipopolysaccharide-induced secretion of IL-1β, IL-8, IL-10, IL-12p40, IL-23, and TNF-α cytokines. Even more importantly in the case of rosacea, the comparison between THP-1 cells cultured in the presence of Am-25-d and cells cultured without the polysaccharide fraction revealed that the presence of Am-25-d contributed to the decrease in the nuclear concentration of the pro-inflammatory nuclear factor kappa NF-kB (which has been shown to have a big impact in the early stages of rosacea) [101].

*Achillea* spp. are traditionally used to treat various skin irritations. An in vivo, double-blind, randomized study analyzed the maceration of the aerial parts of *A. millefolium* in sunflower and olive oil, using 8% (*v*/*v*) sodium lauryl sulfate as an artificial irritant [101]. The study demonstrated that these macerates possess significant anti-inflammatory and soothing properties on the skin. The skin parameters taken into consideration were skin capacitance, pH, and erythema index. All were returned to baseline values after three or seven days of treatment [101]. This study shows that *A. millefolium* extract is a candidate for the management of rosacea-affected skin.

### 5.13. Coffea arabica L. (Coffeeberry)

It is well known that coffee plant (*Coffea arabica* L.) extracts possess antioxidant properties [102]. Coffeeberry contains potent polyphenol compounds, including chlorogenic acid, proanthocyanidins, quinic acid, and ferulic acid [79]. These polyphenols have been proven to offer protection against UVA and UVB radiation and contribute to the prevention of damage caused by oxidative stress and free radical exposure [88].

Yi Fang Li and Shu Hua Ouyang et al. conducted a complex study that investigated the ability of caffeine to prevent oxidative stress-induced senescence and the mechanisms of action. They concluded that even low doses (<10 μM) of caffeine are able to suppress cellular senescence and skin damage through a mediated autophagy mechanism [103].

In a recent 2022 in vitro study, Nisakorn Saewan et al. used a 50% ethanol solution to evaluate the antioxidant properties of *Coffee berry* extract in three different assays: superoxide dismutase (SOD) activity, NO inhibition, and anti-collagenase inhibition. The extract showed SOD and NO inhibition activity superior to caffeine and high anti-collagenase activity [104].

Notably, a recent study found that, when compared with healthy control humans, the main risk factors for rosacea were photosensitive skin types, a positive family history of the condition, or former smoking status. This is noteworthy because oral caffeine consumption had previously been thought to be a risk factor for rosacea activity [22]. The volatile oils found in coffee grains or the additional preservatives and fragrances in topical products are more likely to be the cause of dermatitis and/or allergic reactions described in the literature than the caffeine itself [58].

### 5.14. Aloe vera L.

*Aloe vera* has been studied due to its numerous beneficial effects on the skin. Its extraordinary wound-healing proprieties have been proven throughout many clinical trials [105]. Besides this well-documented effect on the skin, there are also studies that show great results for its anti-inflammatory, analgesic, and antipruritic properties [106,107]. Although the aloe plant consists of more than 99% water content, the remaining solid composition is complex and rich in active compounds such as enzymes, polysaccharides, fat-soluble vitamins, minerals, and flavonoids [108].

There are three important effects of the aloe plant that are taken into consideration for its potential use in rosacea: antioxidant, anti-inflammatory, and antimicrobial effects. Dinesh et al. conducted an in vitro study to evaluate the radioprotective effect of aloe and its possible mechanisms. They concluded that *Aloe vera* L. gel is able to scavenge free radicals and NO depending on the dose that is used [109]. It is also known that *Aloe vera* L. contains anthraquinones, which are structurally analog to tetracyclines. This is why aloe gel acts by inhibiting bacterial protein synthesis, making it hard for bacteria to grow wherever it is present [108].

There have been case reports that noted an improvement in itching, burning, and even scarring associated with UV-induced dermatitis. Syed et al. conducted a placebo-controlled double-blind study that showed a significant clearing of the psoriatic plaques (82.2% vs. 7.7% placebo) when a 0.5% aloe extract hydrophilic cream was applied three times daily [110,111]. Another study conducted on burn-wound rats demonstrated significant decreases in vasodilation and vascular wall permeability in the aloe vera-treated group, suggesting an important role in inflammatory skin conditions [112]. Taking all this into consideration, *Aloe vera* L. extract should make a great candidate for caring and soothing rosacea-affected skin.

## 6. Limitations

The limitations of our study refer to the existence of a small number of clinical and preclinical studies and the difficulty of assessing severity and treatment efficacy. Limited research exists on the extracts and phytocompounds discussed in this review. Newer studies may provide more clarity in the future. For now, we present the latest findings on this topic. Moreover, studying rosacea therapies is challenging due to the lack of standardized methods for assessing severity and treatment efficacy. Many studies only report changes in global rating scales rather than describing symptom improvement. Also, most of the formulations that use botanical extracts are not standardized in any way and have not been clinically tested. We need to address this to ensure proper evaluation and treatment.

## 7. Conclusions

When it comes to treating rosacea, there are still a large number of questions regarding the pathophysiology of the condition. Currently, the global trend is to increase the use of cosmetic preparations based on plant extracts. This review shows that there are a number of active botanical extracts/pure isolated phytocompounds attributed with anti-inflammatory, vasoprotective, antioxidant, and antimicrobial properties (one or more of them) that show great promise for the management of this pathology, especially for the first two primary features (erythema and papules). Some insights into the antiangiogenic mechanism have been also conducted, but these data are still poorly reported.

It can be concluded that most of the existing clinical studies address the resolution of specific symptoms (e.g., anti-inflammatory, antioxidant) rather than the disease itself. Clinical research on botanical agents used to treat rosacea remains a groundbreaking contribution to the field of pharmacognosy. The aim is the development of pharmaceuticals with the ability to reduce the off-target effects and enhance the efficacy and therapeutic compliance of patients diagnosed with rosacea.

This update provides a complete picture of active botanicals and opens new avenues for research on natural compounds in the management of rosacea, alone or in combination with consecrated drugs.

## Figures and Tables

**Figure 1 pharmaceuticals-17-00212-f001:**
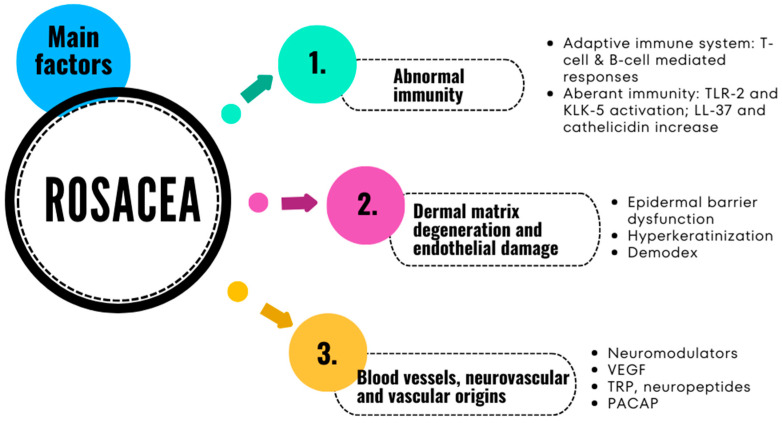
Main factors that contribute to the appearance of rosacea [20,21,22,23,24].

**Figure 2 pharmaceuticals-17-00212-f002:**
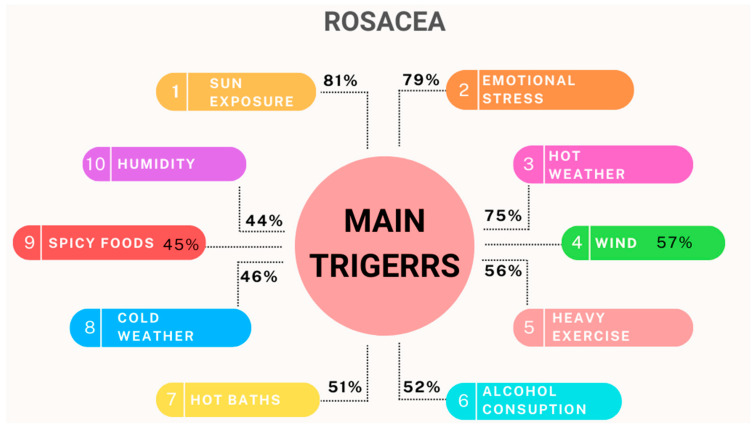
Main triggers of rosacea.

**Figure 3 pharmaceuticals-17-00212-f003:**
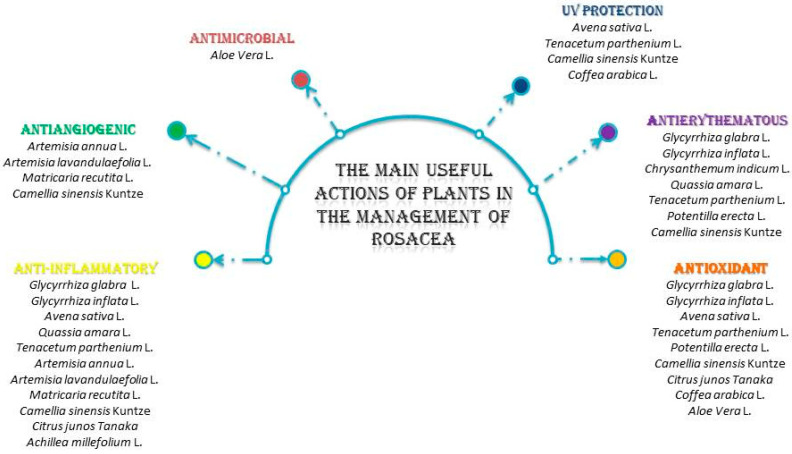
The main useful actions of plants in the management of rosacea.

**Table 1 pharmaceuticals-17-00212-t001:** Major and secondary phenotypes of rosacea [6,7,8,9,10].

**Primary (major) features**	Flushing (erythema)	Redness that is visible and usually prolonged over the central face. It occurs rapidly after being exposed to triggers.
	Papules and pustules	Papules with or without pustules; occasional nodules.
	Telangiectasia	The presence of red blood vessels on the surface of the cheeks. This feature is rarely visible in darker phototypes.
	Ocular manifestations	It may be a feature or a separate form of rosacea altogether. Characterized by lid inflammation or conjunctival redness. In time, it may result in corneal damage.
**Secondary features**	Burning or stinging	Acute sensation of burning or stinging, especially in the central area.
	Dry appearance	Rough and scaly skin, especially around the nose and cheeks area.
	Edema	It is usually present together with erythema but not conditioned by it. It appears when exposed to triggers and may last for a few days.
	Phymatous changes	Thickening or fibrosis of the skin with a bulbous character. The most common form is rhinophyma.

**Table 2 pharmaceuticals-17-00212-t002:** Clinical subtypes of rosacea [8,9,10,11,12].

Clinical Subtypes	Description
Erythemato-telangiectatic (ETR) (subtype I)	Nontransient episodes of flushing; often with stinging and itching; mostly persistent central facial erythema; and telangiectasia is common.
Papulopustular (subtype II)	Persistent facial erythema; papules or pustules or both, with a central facial localization.
Phymatous (subtype III)	Thickened skin, irregular surface nodularities; enlargement of skin tissue. It occurs mostly in the nose area (rhinophyma), but it also affects the chin, forehead, cheeks, or ears. Telangiectasia is also present.
Ocular (subtype IV)	Includes inflammation of the lids and conjunctival telangiectasias. Foreign body sensations in the eye are also common. Patients also complain about dryness, itching, ocular photosensitivity, or even periorbital edema.

**Table 3 pharmaceuticals-17-00212-t003:** Examples of natural compounds used for the management of rosacea.

Product	Source	Active Components
Licorice root	*Glycyrrhiza glabra* L., *Glycyrrhiza inflata* L.	Saponins (glycyrrhizin), flavanones (liquiritin, liquiritigenin), chalcones (isoliquiritigenin, licochalcone), and isoflavones (glabridin, 4t-O-methylglabridin) [39,40].
Indian chrysanthemum flower	* Chrysanthemum indicum * L.	Flavonols (rutin, quercetin), flavones (luteolin), chlorogenic acids, and volatile oils (Germacrene D, α-neoclovene, eucalyptol, and α-pinene) [41,42].
Colloidal oat powder	*Avena sativa* L.	Phenolic acids (caffeic acid, coumaric acid), alkaloids (avenanthramides), unsaturated fatty acids (palmitic, oleic, and linoleic acids), and β-glucan [43].
Amargo wood	* Quassia amara * L.	Steroids (β-sitosterol, stigmasterol), triterpenes, and bitter principles (quassia, neoquassin) [44].
Feverfew leaves	*Tanacetum parthenium* L.	Volatile oils (camphor, *trans*-chrisantenyl acetate) [45], flavones (luteolin, apigenin) [46], and sesquiterpene (parthenolides) [47].
Wormwood plant	* Artemisia annua * L.	Volatile oils (camphor, 1,8-cineole, α-pinene) [48], terpenes (artemisinin), and flavones (apigenin, luteolin, and chrysin) [49,50].
Chamomile flowers	*Matricaria recutita* L.	Chlorogenic acids, phenolic acids (caffeic acid), flavones (apigenin, luteolin), and volatile oils (β-farnesene, farnesol, chamazulene, and α-bisabolol) [51].
Tormentil rhizome/flower	* Potentilla erecta * L.	Flavonols (rutin), flavanones (hesperidin), tannins, triterpenoids (ursolic acid, tormentic acid), and b-sitosterol [52].
Green Tea leaves	*Camellia sinensis* L.	Flavonols (quercetin, kaempferol, and myricitin), catechins (catechin, epigallocatechin, and epigallocatechin gallate), and polysaccharides [53].
Artemisia lavandulaefolia essential oil	*Artemisia lavandulaefolia* L.	Essential oils (β-caryophyllene, cis-chrysanthenol, and camphor) [54] and phenolic acids (3,5-dicaffeoylquinic acid–isochlorogenic acid A, and 4,5-dicaffeoylquinic acid–isochlorogenic acid C) [27].
Yuzu citrus fruit	* Citrus junos * Tanaka	Flavanoles (hesperidin) [55] and essential oils (limonene, a-pinene, b-pinene, and terpinene) [56].
*Achillea* spp. flower	*Achillea millefolium* L.	Flavones (apeginenin, lutelin), flavanols (rutin, kampherol), and essential oils (chamazulene, b-pinene, camphor, and bisabolol) [57].
Coffeeberry beans	*Coffea arabica* L.	Alkaloids (caffeine, trigonelline), chlorogenic acids, and lipids (triglycerides, linoleic acid, and palmitic acid) [58,59].
Aloe vera plant	*Aloe vera* L.	Polysaccharides (acetylated glucomannan, arabinan), proteins (lecitin), vitamins (B1, B2, B6, beta-carotene, and folic acid), and amino acids [60].

## Data Availability

Data sharing is not applicable.

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
