# Peer review of "Recent Advances in the Management of Rosacea through Natural Compounds"

_pharmaceuticals, 2024, doi:10.3390/ph17020212_

Round 1

Reviewer 1 Report

Comments and Suggestions for Authors

Unfortunately, overall, this is a low-quality manuscript for an IF 4.6 journal.

Rosacea has many effective treatment options, and here some natural compounds with very limited evidence are listed as alternatives.

The Introduction is very basic and textbook-like summary of rosacea. The statement of purpose is missing from the end of this section.

Keywords include ones that do not give any hits as not proper terms, such as “antiangionetic") and "antierythematous” and it also makes no sense to have complex keywords such as "rosacea treatment", "treatment of rosacea”, "rosacea pathogenesis" and "physiopathology of rosacea", that are not even within the scope of this review.

The figure “Figure 1. Mechanisms and factors of rosacea” is very much simplified and without any citations.

The only meaningful part is “Table 3. Examples of natural compounds used for the management of rosacea”.

“Figure 2. The main useful actions of plants in the management of rosacea” is created without investigating the evidence behind the listed compounds, therefore it is very much misleading.

Following this, many trials of the listed compounds are just listed with short descriptions, but without a synthesis of the evidence.

Conclusions are also very general, we cannot get any synthetized new information from this paper, about which compounds are effective against rosacea.

Comments on the Quality of English Language

Language should be revised by a native speaker.

Reviewer 2 Report

Comments and Suggestions for Authors

This review raises more questions than answers about the disease rosacea. There are insufficient data on the causative agents and pathophysiological mechanisms by which this disease develops. The authors have analyzed and systematized a huge amount of material on these issues. Ultimately, it is understood that the treatment of rosacea is mainly symptomatic, based on cosmetic products. The components of these preparations contain extracts rich in phytocomplexes, which only relieve the suffering of the sick without attacking the cause.

The present work could serve as a fundamental basis on which to plan and conduct further studies and to search for drugs affecting specifically or non-specifically damaged tissues.

I would recommend that the authors correct the grammar of the English language and also address any technical inaccuracies.

Comments on the Quality of English Language

The English language is required.

Reviewer 3 Report

Comments and Suggestions for Authors

Rosacea is a chronic skin disease that is increasing every year, so I think it is an appropriate topic for a review article.

From Sections 4.1 to 4.16 (pages 4-13), the authors organize sections for each plant and present their benefits. I think it would be a better manuscript if it was divided into sections by major useful actions, as shown in Figure 2, and the efficacy of plants corresponding to each action was introduced.

Reviewer 4 Report

Comments and Suggestions for Authors

The authors reviewed plants and natural products that may potentially be developed for rosacea treatment. The topic is interesting, but the manuscript needs to be substantially revised.

1. In Introduction, Br J Dermatol. 2017 Feb;176(2):431-438. should also be cited as it is the first concensus that proposed phenotype-based diagnosis and classification of rosacea.

2. Reference 2 describe rosacea affects up to 10% of the population. Please provide precise prevalence based on the reference you cite.

3. Table 1. Flushing (erythema) is one of major features. It should not be put in the first line of the table as if it is the subheading of the column.

4. Table 2. Please clarify the abbreviation of erythemato-telangiectatic rosacea is ERT or ETR. 

5. Methods. For the key word, do the authors mean antiangionetic or antiangiogenic?

6. Table 3. Please also provide reference for Artemisia lavandulaefolia essential oil. Also, some references in Table 3 are not cited in the main text. Please also describe main findings of these references in the main text rather than just cite them in the Table.

7. Conclusions. Erythema and papules are phenotypes of rosacea. It is incorrect to describe them as the first two stages of its evolution.

8. Citation format is not unified. For example, reference 2 doesn't mention the journal name. Also, reference 1 and reference 24 cite papers published in the same journal, but one use abbreviation while the other use full name of the journal.

Comments on the Quality of English Language

The reviewer suggests the authors send the manuscript to a native English speaker for English editing. A lot of sentences in the manuscript have unclear meaning. For example, in Abstract, "Thus, the dermato-cosmetic products with natural ingredients are gaining more and more notoriety in front of the, synthetic ones......." It reads as if natural ingredients are notorious in contrast to synthetic ingredients.  

Reviewer 5 Report

Comments and Suggestions for Authors

Your work is interesting and the pathology you have chosen to investigate, rosacea, being a chronic disease, desperately needs new treatments that can be safely administrated indefinitely. 

The strong points of your review are the description of potentially useful NATURAL substances and a rather comprehensively and well organized writing.

Unfortunately, the strong points stop here. The weakest two aspects of your work are:

1. first and foremost, at least to me, your work sounds misleading

(example: "Another study conducted on burn-wound rats demonstrated significant decreases in vasodilation and postcapillary vascular permeability on the aloe vera-treated group, suggesting an important role in inflammatory skin conditions, such as rosacea [96]. " . Yet, the 96 reference refers to a study that evaluated aloe vera ONLY in psoriasis. I have read the whole respective article and the term rosacea does not appear even once in it.)

I would normally reject any article with that sort of problem, but given the fact that your work has tremendous potential and fills a void in the treatment of rosacea, I will not reject it PROVIDING you will more thoroughly select the studies you cite and the information you present.

2. Your work seems, at first glance, to have been written in a hurry. Not only a lot of minor English, grammar and minor spelling mistakes are present, but it also seems you have not fully understand the pathology presented. This observation is evident in Figure 1 when you have combined the supposed pathogenic mechanisms with the trigger factors of rosacea. Please do not do that as you create confusion. The trigger factors should and must be treated separately, as their avoidance represents the first (and until further discoveries, the most important yet) step of treatment.

Other flaws indentified include:

3. The review seems to be a review of the literature and yet you approached your research like a systematic review (but you did not followed the PRISMA guidelines for that type of work). Any comments on that? More importantly, here, at Methods subchapter you should discuss and name exclusion criteria you have used.

4. You have not mentioned and discussed your study's limitations.

5. Also discuss more detailed the current common practice regarding treatment of rosacea and the most common side effects (to isotretinoin, doxicicline, local retinoids, etc). Also, discuss the potential side effects and low-points to plant-based formulations (allergies and the lack of a standardized formulation with all its consequences)

Comments on the Quality of English Language

Minor adjustments, mostly on spelling. Please recheck the manuscript more thoroughly.

Reviewer 6 Report

Comments and Suggestions for Authors

In this manuscript, Similie and co-workers reported “Recent Advances in the Management of Rosacea through Natural Compounds” as a review article. This review is a comprehensive up-to-date report of studies that managed to prove the beneficial effects of different botanicals, that may be useful in the short and long-term management of rosacea-affected skin. This manuscript should be of interest to readers interdisciplinary areas of medicinal researchers. Further development of the vegetal products along with their mechanisms of action responsible for alleviating the clinical symptomatology of the disease, aiding in better comprehending the way plant-based products would be expected.

In summary, I think that this manuscript is appropriate for publication in Pharmaceutics, after the following issues are addressed. 

1.     About introduction: I also think the beneficial effects of different botanicals may be useful in the short and long-term management of rosacea-affected skin. Authors should add the references about the traditional Chinese Medicine which are also derived from natural plants {e.g., Med. Sci. 2014, 2(4), 161-172; Compounds 20222(4), 267-284; Altern. Med. Rev. 2009, 14, 141–153 } .

2.     Authors should align the format in Abstracts. 

3.     If possible, authors should add the representative structures of pharmacological activity from natural plants in Table 3. This information may be helpful for the readers and the medicinal chemists.

4.     Authors should not use general term, such as glycosides, bitter principles, volatile oil, lipid, amino acids, carbohydrate, inorganic compounds, in Table 3. More specific compound names, such as camphor, b-sitosterol, and coumarins, would be required in scientific papers. Authors should rewrite these sections.

Round 2

Reviewer 1 Report

Comments and Suggestions for Authors

Upon a careful review of the revised manuscript, the same issues described in my original review reports are there. Even the new content is of low scientific standard, especially Figure 2, that could be fit for a magazine maybe. 

Comments on the Quality of English Language

Minor issues with the language

Reviewer 4 Report

Comments and Suggestions for Authors

The authors have addressed my previous comments and substantially revised the manuscript. I have no more suggestions.

Reviewer 5 Report

Comments and Suggestions for Authors

Bravo! Your modifications made to the original paper improved a lot the quality of the manuscript.
